# Factors Associated with Levels of Organophosphate Pesticides in Household Dust in Agricultural Communities

**DOI:** 10.3390/ijerph19020862

**Published:** 2022-01-13

**Authors:** Grace Kuiper, Bonnie N. Young, Sherry WeMott, Grant Erlandson, Nayamin Martinez, Jesus Mendoza, Greg Dooley, Casey Quinn, Wande O. Benka-Coker, Sheryl Magzamen

**Affiliations:** 1Department of Environmental and Radiological Health Sciences, Colorado State University, Fort Collins, CO 80523, USA; grace.kuiper@colostate.edu (G.K.); Bonnie.Young@colostate.edu (B.N.Y.); sherry.wemott@colostate.edu (S.W.); grant.erlandson@colostate.edu (G.E.); gregory.dooley@colostate.edu (G.D.); abenkac@gmail.com (W.O.B.-C.); 2Central California Environmental Justice Network, Fresno, CA 93727, USA; nayamin.martinez@ccejn.org (N.M.); xuxomp@gmail.com (J.M.); 3Department of Mechanical Engineering, Colorado State University, Fort Collins, CO 80523, USA; casey.quinn@colostate.edu

**Keywords:** pesticides, exposure modeling, environmental monitoring, empirical/statistical models, environmental justice, vulnerable populations

## Abstract

Organophosphate (OP) pesticides are associated with numerous adverse health outcomes. Pesticide use data are available for California from the Pesticide Use Report (PUR), but household- and individual-level exposure factors have not been fully characterized to support its refinement as an exposure assessment tool. Unique exposure pathways, such as proximity to agricultural operations and direct occupational contact, further complicate pesticide exposure assessment among agricultural communities. We sought to identify influencing factors of pesticide exposure to support future exposure assessment and epidemiological studies. Household dust samples were collected from 28 homes in four California agricultural communities during January and June 2019 and were analyzed for the presence of OPs. Factors influencing household OPs were identified by a data-driven model via best subsets regression. Key factors that impacted dust OP levels included household cooling strategies, secondary occupational exposure to pesticides, and geographic location by community. Although PUR data demonstrate seasonal trends in pesticide application, this study did not identify season as an important factor, suggesting OP persistence in the home. These results will help refine pesticide exposure assessment for future studies and highlight important gaps in the literature, such as our understanding of pesticide degradation in an indoor environment.

## 1. Introduction

Agricultural pesticides can be effective chemical agents to support crop production. However, many commonly used pesticides have high toxicity; even small amounts can adversely impact human and environmental health. Prior studies support a link between pesticide exposure and increased risk of cancer [1,2,3,4], neurodegenerative disease [5,6,7], impaired neurodevelopment in children [8,9,10], and adverse respiratory outcomes, such as asthma morbidity [11,12], chronic obstructive pulmonary disease (COPD) [13], and decreased lung function [14].

California leads the U.S. in agricultural pesticide use, accounting for nearly 20% of total pesticides applied in the U.S. [15]. In 2018, a total of 95 million kg of applied active ingredients were used in California, treating 427,000 cumulative km^2^ [16]. The state began reporting agricultural pesticide use in 1990 with the Pesticide Use Report (PUR), which is considered the most comprehensive program of its kind in the world [16]. The PUR includes self-reporting by applicators of date, location, type, amount of pesticides applied, crop types, and planted acres, plus several other items [16]. Although data are intended to be available annually to represent the previous year’s data, there is a multi-year lag, an important limitation for the PUR to be used as an exposure assessment tool.

The PUR is currently a useful tool to understand spatiotemporal trends in pesticide application. Characterization of residential- and individual-level characteristics that influence pesticide exposure may also support its adaptation as an exposure assessment tool for human health studies. Organophosphates (OPs), for example, are a particularly concerning class of pesticides due to their widespread use in California and their established toxicity [17,18]; however, household characteristics and individual behaviors that are associated with personal exposure to OPs are not well documented [19]. Agricultural communities experience higher exposures and impacts of pesticides compared to non-agricultural areas due to proximity to cropland and orchards (i.e., agricultural fields) that are routinely treated [12,15,20]. In California, where more than 95% of farmworkers are people of color (predominantly Latinx) and lower socioeconomic status [21], these dynamics are also important for environmental justice considerations.

Traditional pesticide spraying and application methods introduce multiple routes of secondary exposure through inhalation, dermal, and ingestion pathways for residents who live close to locations where pesticides are applied. These routes can include residues tracked in on skin, clothes, and shoes of agricultural workers (i.e., para-occupational exposure), and by airborne movement, (i.e., pesticide drift) through dust, droplets, volatilized particles, and soil particles that may enter the home [20,22,23]. Indoor home exposure to pesticides via household dust has been previously identified as an important exposure pathway among agricultural workers and their family members [22,23]. Previous studies showed higher OP dust concentrations in agricultural households and those nearer agricultural fields compared to non-agricultural households or those that were further away [19,24,25].

The objective of this study was to identify household and resident factors that are associated with OPs in household dust among high-exposure counties in the Central Valley of California at two time points representing agricultural non-application (winter) and application (summer) seasons. Unlike technologies that exist for exposure assessment of other environmental and chemical pollutants, the tools that are currently available for measuring pesticides are limited. In addition, heterogeneous exposure pathways complicate exposure assessment among agricultural communities. Our goal is to understand multifactoral influences on pesticide exposure among residents in the Central Valley to support relevant epidemiological studies. We also sought to characterize the temporality of pesticide exposure in the home, given seasonal trends in pesticide application and in light of the potential persistence of OPs in indoor environments. Our findings will help us to determine if a deeper understanding of personal and household influences also provides information about seasonal pesticide exposure. Similar to work that has been done in other agricultural regions of the U.S. [22], our findings may also identify measures that members of agricultural communities can take to mitigate personal exposure and protect their health against the hazardous effects of OPs.

## 2. Materials and Methods

### 2.1. Study Area and Population

The Study of Environmental Mixtures in Periurban Respiratory Outcomes (SEMIPRO) was conducted in Fresno and Tulare Counties in the Central Valley. These counties are ranked first and second in the United States for agricultural products sold [26,27]. Most of the crops produced in Fresno and Tulare are fruits, tree nuts, or berries. According to the PUR, there were approximately 19.8 million kg of pesticides applied in Fresno and Tulare Counties during 2018, which is 21% of total pesticide use statewide. There are approximately 61,000 hired agricultural workers in Fresno and Tulare Counties of which 24,000 and 7000 are migrant and unpaid farm workers, respectively [28]. Furthermore, more than 150,000 Fresno and Tulare Counties residents (>10% of total) live in rural areas that may be exposed to pesticides from nearby spraying and application methods [29].

For this study, four high exposure communities within Fresno and Tulare Counties were selected using 2016 data from the California PUR, which were the most current data available at the time recruitment was conducted in 2018 (Figure 1). According to the PUR, all four of these communities are above the 92nd percentile for the amount of pesticides applied (kg) during 2018 by range level of the Public Lands Survey system among all ranges in California. Central California Environmental Justice Network (CCEJN) staff first mapped the high-exposure communities using Google Maps to identify homes within 61 m (200 feet) of agricultural fields; then, a CCEJN field campaign coordinator implemented a door knocking campaign to enroll eligible households in the study. CCEJN also leveraged existing relationships with other organizations to enroll participants and become active in community-based events to familiarize residents with the project. Roughly 200 households from four agricultural communities were contacted by CCEJN staff through these processes. The four communities that are represented in this study had previously collaborated with CCEJN and had a sufficient availability of households within the requisite 61-m radius of agricultural fields.

### 2.2. Sample Collection and Analysis

Initial study visits were conducted during January 2019 and follow-up visits were conducted during June 2019 to represent the agricultural non-application and application seasons, respectively (Figure 2). During study visits, an adult in the home responded to household- and environment-related survey questions in English or Spanish, based on the participant’s language. The questionnaire was adapted from EPA’s Building Assessment Survey and Evaluation (BASE) [30] to assess indoor air quality. Respondents provided information about household characteristics such as flooring, window coverings, and heating and cooling methods, as well as potential environmental exposures such as the use of pest and weed control chemicals in and around the home. Participants were also asked about their potential exposure to pesticides and dust at work to determine household-level secondary exposure to occupational hazards. Survey data were collected and stored using REDCap electronic data capture tools hosted by the Colorado Clinical & Translational Sciences Institute. All study procedures were approved by the Institutional Review Board at Colorado State University.

OP exposure was assessed by measuring pesticide levels in household dust [31,32,33]. If there was carpeting and/or a rug present in the home, field team members collected household dust samples with a high-volume small surface sampler (CS_3_, Bend, OR) using standard methods as described in the relevant operations manual [34]. Hence, the number of samples collected during each visit was dictated by the availability of carpet or rugs within the surveyed households. After sample collection, household dust samples were stored in a −80 °C freezer at California State University, Fresno. Samples were shipped overnight on dry ice to Colorado State University for analysis, where they were stored at −80 °C until analyzed.

Post-sieving, 100 ± 10 mg of dust were aliquoted from each sample into 2 mL tubes for analysis. The aliquoted mass to the nearest mg was used to calculate the concentration of OPs detected in ng per g of dust (ppb). For 7 samples, less than 100 mg sieved dust was obtained; for these, the entire sample was tested, and its total mass was used for calculations. Samples were analyzed for eleven of the top OPs applied in the study area (by weight) during 2011–2016, as determined from the PUR: acephate, oxydemeton, dimethoate, dibrom, phosmet, malathion, bensulide, diazinon, phorate, chlorpyrifos, and tribufos. Analyses followed methods established in the literature [35] and protocols established in the CSU Analytical Toxicology Laboratory. OPs were extracted from sieved dust samples by sonication in 1 mL acetonitrile for 30 min. The acetonitrile extracts were stored at −30 °C for two hours to precipitate lipids and then centrifuged. Supernatants were further purified by vortexing for 30 s with dispersive solid phase extraction sorbent containing C_18_, graphite, and magnesium sulfate (Agilent Universal d-SPE). Finally, supernatants were transferred to autosampler vials for LC-MS/MS analysis. Calibrators and controls were prepared by spiking household dust that was free of target OPs and extracting with the same method as collected samples.

Prepared calibrators, controls, and samples were analyzed with an Agilent 1290 UHPLC coupled to an Agilent 6460 triple quadruple mass spectrometer equipped with an Agilent Jet Stream electrospray ionization source (Agilent, Santa Clara, CA, USA). Pesticides were first chromagraphically separated on an Agilent Poroshell C_18_ column (2.1 × 100 mm, 2.7 μm) held at 40 °C. A sample volume of 10 μL was injected and a mixture of water with 5 mM ammonium formate/0.05% formic acid (A) and 5 mM ammonium formate/0.05% formic acid in methanol (B) at a flow rate of 0.4 mL/min. The gradient elution used was 10% B for 1 min, 15% B at 1.5 min, 70% B at 2.5 min, and 100% B at 10 min. The ionization source conditions used were as follows: nebulizer 45 psi; gas flow of 12 L/min at 300 °C; sheath gas flow of 12 L/min at 375 °C. The electrospray ionization polarity was set to positive for all analytes. Two ion transitions (m/z) were monitored for each analyte and deuterium labeled internal standards. These ion transitions and corresponding fragmentor and collision energy voltages are displayed in Appendix A. Compound identifications were confirmed by retention time and the product ion ratios (± 20%). The data collection and processing were performed using Agilent MassHunter Quantitative software (v.B.08.01). Quantitation was performed with linear regression using 6-point calibration curves from 5 to 500 ng/g.

### 2.3. Data Management

OP concentrations in the dust were provided in parts per billion (ppb). Each of the eleven OPs assessed was analyzed independently and had a unique limit of quantitation (LOQ). Samples for which an individual OP was not detected were assigned a value of zero for that OP. For samples with detectable OP levels that were below the LOQ a value equal to the LOQ divided by the square root of two was assigned. The sum of all eleven pesticides analyzed in the dust was used to calculate a total OPs in the dust metric. For two samples, initial dust mass was not available, due to recording error. For these samples, if an individual OP was not detected or was detected at a concentration <LOQ, then it was assigned a value using the same method described above. If, however, an individual OP was present in a concentration above the LOQ, then it was assigned a missing value, as was the total OP concentration for those two samples. Prior to modeling, household dust OP concentrations were log-transformed. Log-transformed total OPs, chlorpyrifos, and malathion were used for modeling due to detection in greater than 50% of samples; all other chemicals were not individually assessed due to detection in less than 50% of samples.

### 2.4. Statistical Analysis and Model Approach

A data-driven, best subsets regression approach was employed to evaluate the association of self-reported environmental and occupational exposures and household characteristics on the concentrations of OPs in household dust (Figure 3). Complete lists of the variables included in model building are included in Appendix A. The approach described here was adapted from model-building methods that have been previously described [36] and was applied to identify factors that influence total OPs, chlorpyrifos, and malathion in household dust.

First, univariable models were built for each covariate. A *p*-value cutoff of 0.25 was used to identify covariates that were potentially associated with the outcome of interest (total OPs, chlorpyrifos, or malathion concentrations in household dust). For multilevel variables that were assessed using multiple choice or multiple option survey questions (i.e., “choose all that apply”), all levels passed to the next step of the model building approach if at least one met the 0.25 *p*-value cutoff. For example, participants could indicate that their home had blinds, curtains, and/or shades as window coverings; if the presence of blinds, curtains, or shades in the home had a *p*-value less than 0.25 in a univariable model with the outcome of interest, all three window covering options were included in the next step of the model building approach.

Next, covariates that had a *p*-value of less than 0.25 in the univariable model were evaluated for near-zero variance. Near-zero variance variables were those that met one of two criteria: (1) the ratio of most common to second-most common value was greater than 95:5, or (2) the percentage of unique values out of the total number of observations was greater than 10%. Covariates that met at least one of these two criteria were excluded from further consideration.

All remaining covariates were included in a best subsets regression analysis, and the set of variables selected by best subsets would be included in a single consensus model. An exhaustive best subsets approach was used; covariates were selected for the best possible fit of models that increased in size by one degree of freedom at a time. Best subsets regression was halted when the unique set of variables that had been selected, along with several a priori covariates, would build consensus models of a predetermined size. The number of degrees of freedom allowed in the consensus model was 10, and the following variables were included in the consensus model a priori: community, the presence of carpet in the home, visit month, and secondary occupational exposure to pesticides/agriculture (i.e., at least one household member had self-reported exposure to pesticides/agriculture at work).

The set of variables selected by best subsets and the a priori variables were fit in a full, consensus model that also included a random effect to account for repeated measures at the household level. Here, only the specific levels of multilevel variables that were selected by best subsets were included. For example, if the presence of blinds in the home was selected by the best subsets regression, but the presence of shades or curtains were not selected, then only the presence of blinds in the home was included in the consensus model.

Finally, the consensus model was further reduced in a stepwise manner in which the variable with the highest *p*-value was eliminated. If the Akaike Information Criterion (AIC) of the reduced model was not smaller than the previous and/or if there was confounding by the eliminated variable, as evidenced by a >20% change in beta estimate of other covariates, the eliminated variable was kept in the model. This was repeated until all covariates in the model were identified as confounders, improved model AIC, or were significantly contributing to the model (*p* < 0.05).

All data cleaning and statistical analyses were conducted in R v.4.0.2 [37], primarily using the *tidyverse*, *lme4*, *caret*, and *leaps* packages.

### 2.5. Sensitivity Anlaysis

To test the robustness of the final models, a sensitivity analysis was conducted using only data from Community #1, from which the most samples were collected, and the highest total OP levels were detected. For each of the household dust OP measures that were modeled—total OPs, chlorpyrifos, and malathion concentrations—the final models that were constructed using the best subsets regression approach were fit again, using only data collected from households located in Community #1.

## 3. Results

### 3.1. Descriptive Findings

Surveys were conducted with 34 households during the January visit; only one household was lost to follow-up during the June visit. A total of 28 households were eligible to provide a dust sample during at least one of the study visits. In total, 52 dust samples were collected, 50 of which had sufficient mass for laboratory analysis. The average household occupancy of the 28 sampled households from which dust samples were analyzed for the presence of OPs was 4.0 (standard deviation, SD = 1.5, Table 1). Secondary occupational exposure to dust and agriculture/pesticides was reported among 75% and 71% of households, respectively. Most households (71.4%) used air conditioning (AC) to cool the home; other common cooling strategies included window cooling units, open windows, and fans/house fans (used by 25.0%, 25.0%, and 7.1% of households, respectively).

Household dust OP concentrations were quantified for 50 dust samples (Table 2, Figure 4). OPs were detected in most samples (93.6%); chlorpyrifos and malathion were detected in 79% and 76%, respectively. Total OP concentrations among household dust samples ranged from 0 to 651.7 ppb. Chlorpyrifos and malathion concentrations in household dust samples ranged from 0 to 395.3 and 0 to 29.1 ppb, respectively. The average total OP concentration of dust samples collected from households in which at least one member had occupational exposure to pesticides was 127.3 ppb (*n* = 33); the average total OP concentration in households in which nobody worked in agriculture was 27.1 ppb (*n* = 14, Wilcoxon rank-sum test: *p*-value = 0.002).

No difference was observed between paired household dust samples by application season for chlorpyrifos (paired Wilcoxon rank-sum test: *p*-value = 0.89), malathion (*p*-value = 0.73), or total OPs (*p*-value = 0.67). Among households that were sampled during both the winter and summer seasons (*n* = 20 households), the average total OP concentration was 99.6 ppb during January and 86.9 ppb during June. The average total OP concentration among all dust samples collected during January was 90.7 ppb (*n* = 23); during June, the average total OP concentration was 103.9 ppb (*n* = 24).

For each individual OP analyzed, Kruskal–Wallis tests were performed to test for statistically significant differences in household dust OP concentrations across the four agricultural communities that were sampled (Appendix A). These tests revealed that acephate, chlorpyrifos, dimethoate, malathion, tribufos, and total OPs levels in household dust samples did differ significantly between communities. Significant differences were not observed, however, for bensulide (not detected), diazinon, dibrom (not detected), oxydemeton-methyl, phorate (not detected), or phosmet.

### 3.2. Total OPs

The variables that were selected by the best subsets model-building approach to be included in the final model for total OPs in household dust are presented in Table 3 (adjusted R^2^ = 0.50). The use of fans/house fans for cooling was found to be significantly associated with total OPs in the dust; household dust collected from homes in which fans were used for cooling had a 2.81 (95% confidence interval (CI): 1.16, 4.47) unit increase in log-transformed total OPs in the dust compared to homes in which fans were not used for cooling. Households in which at least one member worked in agriculture had a 1.07 (95% CI: 0.24, 1.90) unit increase in log-transformed total OPs in the dust compared to households in which no members worked in agriculture. Some geographical differences were observed as well. Community #2 had significantly lower OP levels in household dust samples compared to Community #1.

### 3.3. Chlorpyrifos

Household and behavioral characteristics associated with chlorpyrifos in household dust are presented in Table 3 (adjusted R^2^ = 0.28). Only secondary occupational exposure was found to have an association with the outcome for which the 95% CI did not include the null (β: 1.59, 95% CI: 0.47, 2.70).

### 3.4. Malathion

Household/behavioral characteristics associated with malathion are presented in Table 3 (adjusted R^2^ = 0.38). The use of fans/house fans for cooling was identified as having a significant influence on levels of malathion in household dust (β: 1.19, 95% CI: 0.46, 1.92). Malathion concentrations in household dust changed significantly geographically, as well. Community #2 had significantly lower malathion levels compared to Community #1, whereas higher levels of malathion were detected in dust samples collected from households in Communities #3 and #4. Other variables that were included in the final model for malathion concentrations in household dust were the type of home and the use of window units for cooling; however, these confidence intervals include the null value.

### 3.5. Sensitivity Anlaysis

Community #1 had significantly higher household dust concentrations of 171.4 ppb (SD = 199.5) of total OPs, as compared to 51.6 ppb (SD = 51.6 ppb, *n* = 29) in the other three communities (Wilcoxon rank-sum test: *p*-value = 0.002). Furthermore, the most dust samples were collected from Community #1 (*n* = 18 for total OPs, *n* = 19 for chlorpyrifos, and *n* = 21 for malathion). However, sample size was still limiting for a sensitivity analysis using only Community #1 data and CIs were wide as a result (Appendix A). For the chlorpyrifos and malathion models, the CIs surrounding all estimates included the null value. For total OPs, cooling using fans remained significant (β: 2.40, 95% CI: 0.42, 4.39); the 95% CI around the estimate for secondary occupational exposure to pesticides, though, contained the null. Unlike in the model fit for the primary analysis, having blinds in the home was significantly associated with a decrease in household dust total OP concentrations when Community #1 was modeled alone (β: −1.32, 95% CI: −2.45, −0.19).

## 4. Discussion

Exposure to agricultural pesticides has established links to numerous adverse health outcomes among children and adults. California’s Central Valley has an especially high proportion of pesticide use and people living near agricultural fields, including agricultural workers and their families, who experience various pathways of exposure. One important pathway that has been previously identified is indoor pesticide exposure from household dust, which can be tracked into the home on skin, clothes, and shoes, or blown in from airborne movement [22,23]. Apart from proximity of the household to agricultural fields [19,24,25], individual and household characteristics resulting in OP exposure are less established. We collected household and resident data from Central Valley households within 61 m of agricultural fields at two time points to examine factors associated with OPs in household dust during non-application (winter) and application (summer) seasons.

Overall, we identified three key factors associated with OPs found in household dust. First, consistent with previous studies, we found that secondary occupational exposure to pesticides was significantly associated with chlorpyrifos and total OP concentrations in household dust. Substantial literature supports that agricultural workers serve as vectors for pesticide exposure via the “take-home” route [20,35,38]. Fenske et al. identified strong correlations between agricultural workers’ vehicle dust OP concentrations and the OP levels in their household dust; moreover, vehicle dust OP levels were shown to be higher than those of household dust [22]. This study supports these findings and highlights the importance of investigating the effectiveness of measures that agricultural workers can take to limit their take-home exposure.

Second, cooling methods used within the home significantly impacted malathion and total OP household dust levels. The influence of cooling strategies on household dust OP levels has been previously reported. Harnly et al. and Butler-Dawson et al. have demonstrated significantly lower OP levels in the dust of homes that had AC compared to those that did not [19,39]. Previously, it has been hypothesized that AC users have less need to open windows for ventilation and cooling as compared to those who do not use AC, which may protect their indoor environment from pesticide infiltration [19]. However, opening windows during pesticide application season is unpopular in this study area as the local climate is extremely hot during the summer. Furthermore, participants anecdotally reported keeping windows closed during times when pesticide spraying may occur. Behavioral and climatological differences may explain the lack of significance associated with AC use found in this study as compared to previous findings. The studies conducted by Harnly et al. and Butler-Dawson et al., for example, were located in the Salinas Valley of California and the Pacific Northwest, respectively, which are more moderate during the summers than the Central Valley [19,39]. Our results, however, showed elevated household dust OP levels associated with the use of fans or house fans for cooling the home. These findings still support the consensus that some cooling methods provide greater protection against the infiltration of airborne pesticides into the indoor environment than others.

Lastly, we found that the geographic location by community significantly impacted malathion and total OP concentrations, which may suggest that spatial variation in agricultural production and pesticide application in Fresno and Tulare Counties impacts the quantities and types of pesticides that collect in household dust. Indeed, a sensitivity analysis using only Community #1 samples revealed different associations between the household characteristics selected by best subsets regression and dust total OP levels than the primary analysis. However, these results should be interpreted with caution, as sample sizes were extremely limited when only Community #1 was included. Visualization of household dust pesticide levels also suggests that there is variation in individual OP concentrations across communities (Figure 4). Future studies with larger sample sizes should investigate these potential differences. The PUR is spatially resolved at the county, meridian, township, range, section (CMTRS) level of the Public Lands Survey system; future research should target the validation of this tool to capture spatial variability in pesticide exposure at the community level.

An important takeaway from this study was the null association between season and household dust OP levels. For total OPs, chlorpyrifos, and malathion, visit month did not meet the first criterium of the model building approach (*p*-value > 0.25 in a univariable model). However, visit month was considered a priori to be an important covariate and was therefore included in consensus models (Figure 3). Still, it was eliminated from each final model during stepwise model reduction (Appendix A). Previous reports show that concentrations of OPs in household dust are higher during the thinning and harvest seasons than the non-spraying (December through February) season [23,35]. Our null findings for seasonal differences may suggest long-term persistence of OP pesticides in indoor environments. The limited exposure of pesticides to light, water, and microbial degradation within household air and dust may drastically increase their half-life. For example, in a 5-week pesticide degradation study conducted in the U.S. EPA Indoor Air Quality Test House, Starr et al. reported the detection of chlorpyrifos within the test house, although it had not been applied for several years [40]. Recently, Oudejans et al. found that pesticide degradation took significantly longer in a dark indoor environment—with no meaningful changes prior to 140 days—compared to previously reported half-lives characteristic of soil or water environments [41]. The hypothesized persistence of pesticides may limit the effectiveness of the PUR as an exposure assessment tool if reported seasonal and long-term trends in pesticide application are not reflected in the indoor environment.

During the June visit, we asked participants, “This study is about pesticides, but we are interested in other concerns you may have. So, what other concerns do you and your family have about your environment?” In response to this question, 70% of adults still voiced concerns about pesticides and spraying applications, and 48% also expressed specific concerns about the health of their family, friends, and/or community. Participants expressed feelings of helplessness to protect themselves and their families from exposure to harmful pesticides. Advocacy for these community members should empower them to minimize their own exposures to pesticides. The desire to combat environmental injustice by increasing agency, education, and opportunity among agricultural communities to mitigate pesticide exposures was an important motivation of this study. These results may inform interventions that agricultural workers and their workplaces can take, such as removing contaminated clothes and shoes prior to leaving the agricultural fields, regularly cleaning inside commuter vehicles, and making laundry facilities available at the workplace to decrease secondary occupational exposure to pesticides [22]. It may also be appropriate to recommend different household cooling strategies, such as the use of AC if available, to lower both individual- and household-level exposures.

The identification of factors that influence pesticide exposures among agricultural communities will also serve to increase the precision of exposure assessment for future studies. Current methods for measuring personal pesticide exposure, mainly via biomarkers, are expensive and burdensome for participants. Our results contribute to a more complete understanding of the relevant factors that can influence personal exposure to pesticides. Together, the three unique factors that were significantly associated with chlorpyrifos and malathion levels in household dust (secondary occupational exposure to pesticides for chlorpyrifos, and community and the use of fans for malathion) were the same three factors that were significantly associated with total OP concentrations. This pattern may reflect the substantial contributions of malathion and chlorpyrifos to the overall OP profiles detected among household dust in these agricultural communities. However, the influence of other individual OPs cannot be ruled out, and future research is necessary to fully characterize pesticide exposure via household dust to support the refinement of other available tools, such as the PUR, for estimating household- or community-level exposures. Furthermore, our results reveal a need to investigate the degradation of OP pesticides indoors. Currently there is limited research to understand this phenomenon, despite the prevailing hypothesis that pesticides can persist for long periods of time indoors [41] and the detection of pesticides within indoor environments that implicates historic exposure [40,42,43,44].

This study has several notable strengths. We worked closely with our community partner, CCEJN, to identify high-risk communities for agricultural pesticide exposure due to their close proximity to agricultural fields. We conducted in-person interviews in Spanish and English to gather detailed information on household and resident behaviors. Dust was collected directly from households to estimate agricultural pesticide levels currently present in participant’s homes as a “real-time” assessment of potential exposure.

There were several limitations of this study that are important to note. Firstly, this study had a small sample size, which can increase the uncertainty around model estimates. Although 34 households were recruited to participate in the study, only 28 provided a dust sample given their availability of a rug or carpet for dust collection. Another possible limitation of this study is failure to measure unknown factors. We attempted to capture relevant household characteristics and resident behaviors that may be associated with pesticide exposure; however, participants were not asked to report potential mitigating efforts that they use against pesticide exposure, such as removal of contaminated clothing prior to entering the home or closing windows and doors during pesticide application.

There is a lack of studies that offer real-time OP exposure assessment linked to health outcomes or indicators of health risks. Currently available pesticide exposure assessment tools have major temporal limitations and lack precision; at the time of this publication, data that are relevant to our study period are still unavailable from the PUR. Additionally, tools to measure real-time pesticide exposure are typically burdensome and expensive, such as biomarkers or household dust samples. Therefore, refinement of individual-level pesticide exposure models that are current and able to account for temporal trends is needed for epidemiological studies. Identifying key household characteristics and individual behaviors that influence OP exposures can help relieve participant burden for future longitudinal studies of sub-clinical markers of health. Further, this research can support the health of agricultural community members who experience environmental injustice by clarifying points of intervention to mitigate exposure pathways.

## 5. Conclusions

A data-driven model via best subsets regression was used to identify important characteristics that were associated with household dust OP pesticide levels among four California agricultural communities. The results of this analysis revealed the influence of household cooling strategies and geographic location on concentrations of malathion and total OPs in household dust samples. Secondary occupational exposure to pesticides was found to significantly increase levels of chlorpyrifos and total OPs among sampled households. These findings will support the refinement of pesticide exposure assessment for future studies, including epidemiological analyses to characterize the health risks of OP exposure. Additionally, this study identified null associations between agricultural application season and household dust OP levels, contrary to findings from similar studies. It is possible that the absence of detectable differences in OP levels between the winter and summer sampling periods (i.e., agricultural non-application and application seasons) indicates long-term persistence of pesticides in the household environment. These findings also uncovered an important gap in the literature related to our understanding of pesticide degradation, particularly indoors. Finally, our results may provide insight on risk factors for household OP exposures among agricultural communities that experience environmental injustice related to pesticide exposure and highlight important targets for mitigation strategies.

## Figures and Tables

**Figure 1 ijerph-19-00862-f001:**
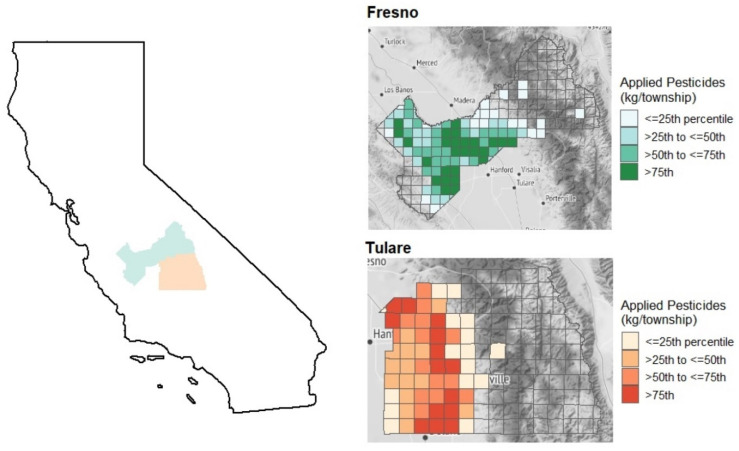
Map of study areas: Fresno (green) and Tulare (red) counties, CA. Identification of high pesticide exposure by county California Pesticide Use Report data, 2018.

**Figure 2 ijerph-19-00862-f002:**
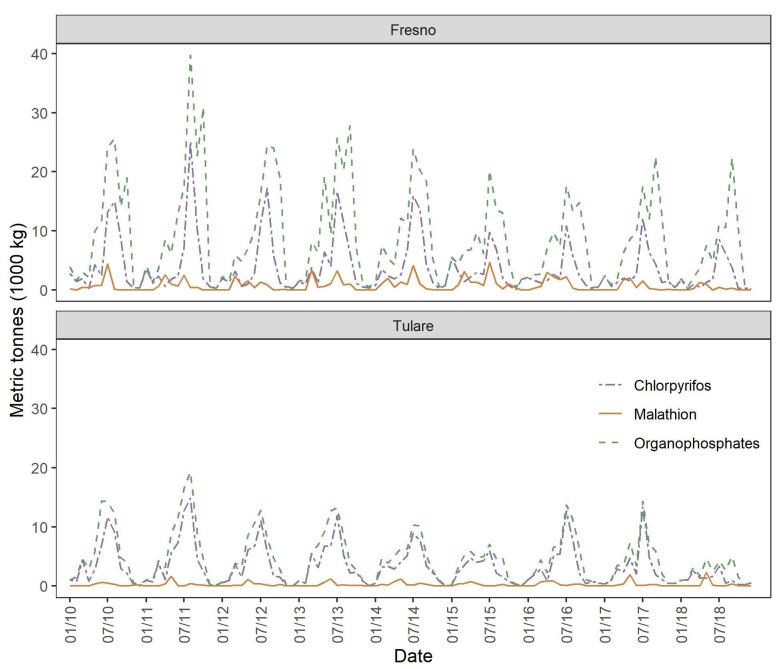
Annual agricultural application (in metric tonnes) by month of total organophosphates in study counties, 2010–2018.

**Figure 3 ijerph-19-00862-f003:**
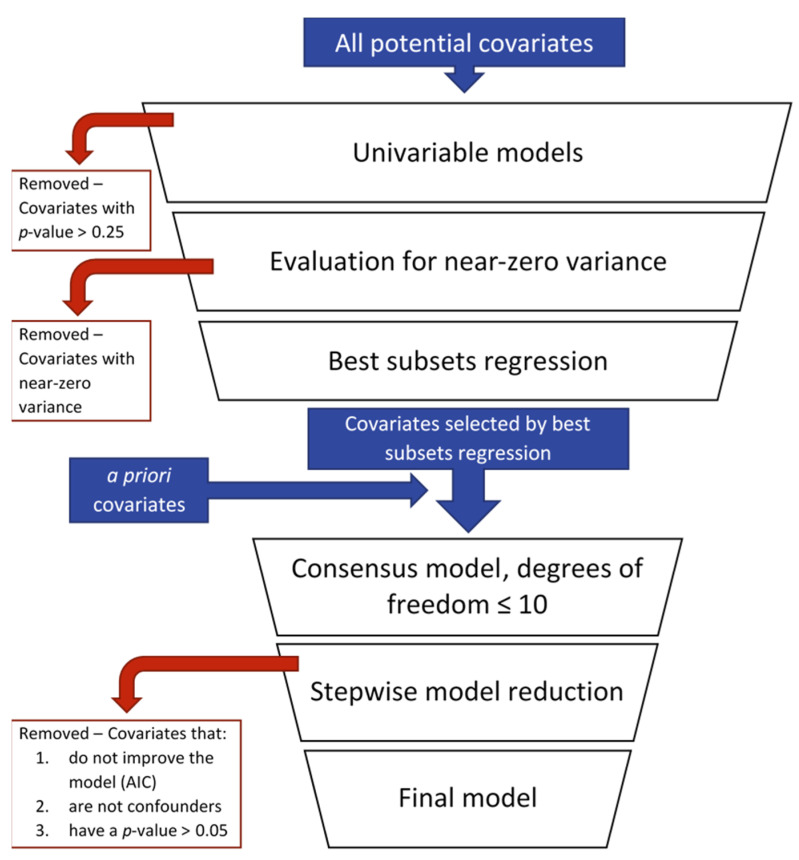
Flowchart of data-driven, best subsets regression approach for model building.

**Figure 4 ijerph-19-00862-f004:**
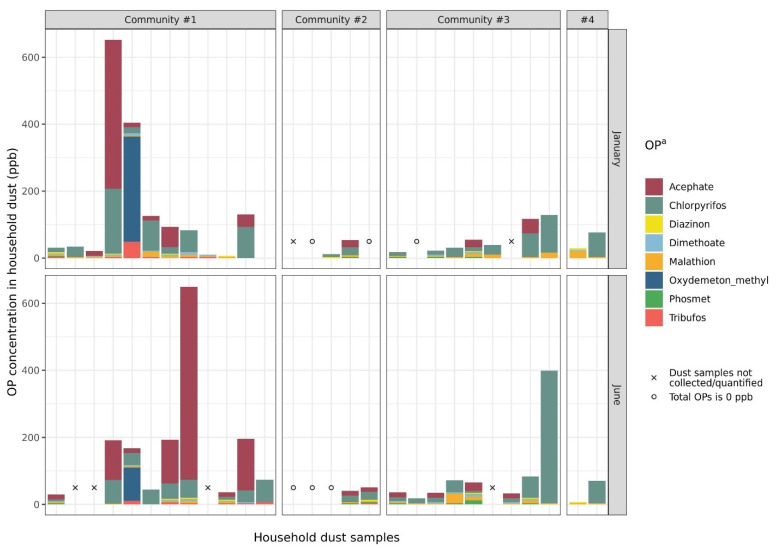
Composition of household dust samples, by sampling season and community. ^a^ Bensulide, dibrom, and phorate were also analyzed; however, these three OPs were not detected in any of the household dust samples.

**Table 1 ijerph-19-00862-t001:** Characteristics of households from which dust samples were analyzed for the presence of OPs.

	Community #1	Community #2	Community #3	Community #4	Overall
	*n* = 12	*n* = 5	*n* = 9	*n* = 2	*n* = 28
Home type, *n* (%)					
House	7 (58.3)	5 (100)	9 (100)	1 (50.0)	22 (78.6)
Other (mobile home or apartment)	5 (41.7)	0 (0)	0 (0)	1 (50.0)	6 (21.4)
Number of rooms, mean (SD)	6.3 (0.9)	5.0 (2.4)	6.4 (2.2)	6.0 (1.4)	6.1 (1.7)
Number of doors, mean (SD)	5.9 (1.7)	9.0 (2.6)	3.3 (1.9)	1.5 (0.7)	5.3 (2.9)
Number of windows, mean (SD)	7.4 (1.5)	7.2 (1.5)	8.9 (2.3)	7.0 (0)	7.8 (1.8)
Home ownership, *n* (%)					
Own	6 (50.0)	4 (80.0)	7 (77.8)	1 (50.0)	18 (64.3)
Other (rent or owned by employer)	6 (50.0)	1 (20.0)	2 (22.2)	1 (50.0)	10 (35.7)
Flooring, *n* (%)					
Carpet	7 (58.3)	5 (100)	7 (77.8)	2 (100)	21 (75.0)
Rugs	9 (75.0)	4 (80.0)	5 (55.6)	0 (0)	18 (64.3)
Window coverings, *n* (%)					
Blinds	6 (50.0)	2 (40.0)	4 (44.4)	2 (100)	14 (50.0)
Curtains	10 (83.3)	5 (100)	7 (77.8)	2 (100)	24 (85.7)
Shades	1 (8.3)	0 (0)	2 (22.2)	0 (0)	3 (10.7)
Warm-blooded pet(s), *n* (%)	7 (58.3)	5 (100)	2 (22.2)	2 (100)	16 (57.1)
Flea/tick treatments for warm-blooded pets, *n* (%)	4 (33.3)	4 (80.0)	0 (0)	0 (0)	8 (28.6)
Heating sources, *n* (%)					
Clean (only electric sources used)	8 (66.7)	1 (20.0)	5 (55.6)	1 (50.0)	15 (53.6)
Dirty (gas and/or oil sources used)	2 (16.7)	3 (60.0)	4 (44.4)	1 (50.0)	10 (35.7)
None (no heating sources used)	2 (16.7)	1 (20.0)	0 (0)	0 (0)	3 (10.7)
Cooling, *n* (%)					
AC	6 (50.0)	4 (80.0)	8 (88.9)	2 (100)	20 (71.4)
Window units	5 (41.7)	0 (0)	2 (22.2)	0 (0)	7 (25.0)
Open windows	4 (33.3)	1 (20.0)	2 (22.2)	0 (0)	7 (25.0)
Fans/house fans	1 (8.3)	1 (20.0)	0 (0)	0 (0)	2 (7.1)
Household chemicals to control ants/bugs	2 (16.7)	4 (80.0)	4 (44.4)	1 (50.0)	11 (39.3)
Household chemicals to control mice/rodents	2 (16.7)	2 (40.0)	1 (11.1)	0 (0)	5 (17.9)
Household chemicals to control weeds	4 (33.3)	3 (60.0)	4 (44.4)	2 (100)	13 (46.4)
Household chemicals to control pests	0 (0)	1 (20.0)	0 (0)	0 (0)	1 (3.6)
Number of adults living in the household, mean (SD)	2.8 (1.1)	4.2 (1.8)	2.4 (0.9)	3.5 (0.7)	3.0 (1.3)
Number of children living in the household, mean (SD)	1.3 (1.5)	1.0 (1.7)	0.9 (0.8)	0 (0)	1.0 (1.3)
Household occupancy, mean (SD)	4.0 (1.5)	5.2 (1.3)	3.3 (1.3)	3.5 (0.7)	4.0 (1.5)
Reported secondary occupational exposure to, *n* (%):					
agriculture/pesticides	11 (91.7)	3 (60.0)	5 (55.6)	1 (50.0)	20 (71.4)
dust	10 (83.3)	4 (80.0)	6 (66.7)	1 (50.0)	21 (75.0)
Household secondhand smoke exposure	1 (8.3)	0 (0)	1 (11.1)	0 (0)	2 (7.1)

**Table 2 ijerph-19-00862-t002:** Household Dust OP Concentrations (ppb).

Exposure Variable	Visit Month	*n*	LOQ	Mean (SD)	IQR	Maximum ^a^	CV (%)	Rate of Detection
				*x* (SD)				% (*n*_detected_)
Acephate	January	24	20	28.0 (90.2)	16.1	444.2	322.1	37.5 (9)
	June	24	20	46.5 (120.9)	14.1	575.4	260.0	54.2 (13)
Bensulide	January	26	5	0 (0)	0	0	-	0 (0)
	June	24	5	0 (0)	0	0	-	0 (0)
Chlorpyrifos	January	24	5	37.7 (47.6)	60.6	193.3	126.3	75.0 (18)
	June	24	5	42.8 (78.8)	40.2	395.3	184.1	83.3 (20)
Diazinon	January	26	5	1.0 (1.8)	2.7	5.7	172.8	26.9 (7)
	June	24	5	1.0 (1.6)	3.5	3.5	159.2	29.2 (7)
Dibrom	January	26	20	0 (0)	0	0	-	0 (0)
	June	24	20	0 (0)	0	0	-	0 (0)
Dimethoate	January	26	5	1.8 (2.3)	3.5	7.2	129.0	42.3 (11)
	June	24	5	2.0 (2.3)	3.5	7.9	112.3	50.0 (12)
Malathion	January	26	5	5.2 (5.4)	2.9	20.5	102.9	76.9 (20)
	June	24	5	4.4 (5.9)	1.4	29.1	135.6	75.0 (18)
Oxydemeton-methyl	January	26	10	12.1 (61.5)	0	313.6	508.3	3.8 (1)
	June	24	10	4.1 (20.2)	0	99.2	489.1	4.2 (1)
Phorate	January	26	10	0 (0)	0	0	-	0 (0)
	June	24	10	0 (0)	0	0	-	0 (0)
Phosmet	January	26	5	0.7 (1.4)	0	3.5	208.8	19.2 (5)
	June	24	5	1.4 (2.8)	3.5	12.4	200.7	29.2 (7)
Tribufos	January	25	5	2.7 (9.7)	0	48.9	365.8	24.0 (6)
	June	24	5	1.6 (3.0)	3.5	11.1	183.3	29.2 (7)
Total OPs	January	23		90.7 (149.3)	85.3	651.7	164.6	100 (23)
	June	24		103.9 (147.7)	72.7	649.2	142.2	87.5 (21)

^a^ The minimum value for each individual OP pesticide was 0 ppb for both seasons.

**Table 3 ijerph-19-00862-t003:** Final household dust models.

Variable	Estimate	95% CI	
		Lower	Upper
*Total OPs (n = 47)*			
Window coverings: blinds ^b^	−0.55	−1.38	0.27
Heating			
Clean (only electric sources used) ^a^	-	-	-
Dirty (gas and/or oil sources used)	0.44	−0.52	1.39
None (no heating sources used)	0.88	−0.44	2.20
**Cooling: fans/house fans ^b^**	**2.81**	**1.16**	**4.47**
**Secondary occupational exposure to agriculture/pesticides ^b^**	**1.07**	**0.24**	**1.90**
Flooring: carpet ^b^	−0.56	−1.79	0.66
Community			
Community #1 ^a^	-	-	-
** Community #2**	**−3.11**	**−4.22**	**−1.99**
Community #3	−0.05	−1.00	0.90
Community #4	0.22	−1.40	1.85
*Chlorpyrifos (n = 48)*			
Window coverings: blinds ^b^	−0.71	−1.84	0.42
Window coverings: shades ^b^	0.74	−1.22	2.70
Number of rooms ^b^	0.11	−0.22	0.44
**Secondary occupational exposure to agriculture/pesticides ^b^**	**1.59**	**0.47**	**2.70**
Flooring: carpet ^b^	−0.65	−2.06	0.75
Community			
Community #1 ^a^	-	-	-
Community #2	−1.22	−2.68	0.24
Community #3	0.45	−0.75	1.65
Community #4	0.31	−1.79	2.41
*Malathion (n = 50)*			
Home type			
House	0.41	−0.14	0.95
Other (mobile home or apartment) ^a^	-	-	-
Cooling: window units ^b^	0.46	−0.0002	0.93
**Cooling: fans/house fans ^b^**	**1.19**	**0.46**	**1.92**
Community			
Community #1 ^a^	-	-	-
** Community #2**	**−0.85**	**−1.44**	**−0.25**
** Community #3**	**0.60**	**0.10**	**1.11**
** Community #4**	**0.98**	**0.23**	**1.72**

^a^ Indicates referent level for indicator or categorical variables. ^b^ For indicator variables, the referent is the absence of the covariate/exposure in the home (e.g., the referent for ‘window coverings: blinds‘ is not having blinds in the home). Bold typeface indicates variables that were significant as determined by a 95% CI that did not include the null.

## Data Availability

Survey instrument and all code are available upon request from Grace Kuiper. Due to IRB considerations, we are prohibited from sharing household data collected as part of this study.

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
