# Peer review of "Factors Associated with Levels of Organophosphate Pesticides in Household Dust in Agricultural Communities"

_ijerph, 2022, doi:10.3390/ijerph19020862_

Round 1

Reviewer 1 Report

The manuscript by Kuiper et al., “Factors associated with levels of organophosphate pesticides in household dust in agricultural communities”, focuses on the identification, by means of a statistical model, of household and resident factors associated with organophosphate pesticides. Household dust samples were collected from 28 homes, within 61 m of agricultural fields, in California agricultural communities during January and June 2019.

The topic is of interest for the scientific community; it is an interesting study that includes an important dataset. However, this manuscript has important weaknesses that should be addressed before accepting the manuscript for publication:

  • There are already important studies related to this topic with similar conclusions. In fact, in lines 299-323, authors mention other studies with similar results. A deeper analysis of the results should be carried out. For example, in lines 299-323, it is not enough to say that there are studies that have found similar results, it is necessary to hypothesize about the main cause/s of those results.
  • Statistical analysis and model approach: this section is crucial for the rest of the paper and it is not clear to me from the text. Authors should try to give a better explanation of the methodology applied; maybe a scheme/diagram would be helpful.
  • Fig. 3 and Table 2 provide information about OPs concentrations (by sampling season and community and mean concentrations, respectively). However, authors have not provided an analysis of these individual concentrations. Even if it is not the main objective of the work, I think that it could be an important point for enriching the work.
  • Fig. 3: What could be the reasons for the important differences found among the concentration registered for the different communities? Can we join the data from the four communities and work as if they were one group? Are there statistically significant differences among the OPs concentrations registered in the four communities? A statistical test should be applied.
  • Would it be possible to establish a model for the community 1 data (it is the community with more samples)? It is a topic that should be explored in the manuscript.
  • Table 2: The standard deviation for several OPs are really important (28± 90.2, 12.1± 61.5, 2.7±9.7, 4.1± 20.2, 46.5± 120.9 ppb, etc.). Authors should discuss the consequences of this variability for the main results of the study.
  • Authors point out a null association between season and household dust OPs levels. Has this analysis been performed for each of the OP tested or for the total OPs concentration? Please, indicate the data considered for the analysis. How has the analysis been conducted? It is important to clarify this fact in the manuscript.
  • Are Fresno and Tulare data comparable?
  • I propose to join the discussion and conclusion sections, trying to emphasize the main novelties of the work, avoiding repetitions and reflections that do not derive directly from the scientific study carried out.

Minor comments:

Lines 55-56: I wonder if the skin colour is a relevant information for this scientific paper.

Lines 77-78: Please, clarify this sentence.

Lines 80-83: There are other papers that have already identified these measurements, for example, Fenske et al. (2013)

Lines 246-250: Authors should indicate the number of samples used for the calculation of the mean concentrations and specify the values for the two periods studied.

Line 254: Separate “any” from “of”

Lines 260-263: Which could be the main reason/s?

Table 1: Indicate the meaning of “SHS” and eliminate “n(%)” from the last row.

Table 2:

  • I propose to place the mean concentrations recorded during the two periods studied side by side in order to be more easily comparable. As the minimum value is always zero, it is not necessary to include this information in the table (it could be said in the heading of the table). If necessary, information regarding IQR, maximum, CV and rate of detection could be included in the supplementary material.
  • Indicate the meaning of “CV” and “IQR”

Tabla 3:  

  • Authors should discuss the differences found for Total OPs, Chlorpyrifos and Malathion.
  • Several non-significant variables have been included in the table; why?
  • Please, draw a horizontal line for separating the different models

What is the influence of meteorological conditions on the transport of pesticides to homes? I think that it is an important point to consider for future studies.

Reviewer 2 Report

ijerph-1490144-peer-review-v1

The authors analyze factors influencing OP pesticide concentration in house dust in households near agricultural areas with high pesticide application.

This approach is highly recommendable. But a set of only 28 households is maybe not sufficient to make strong predictions. Nevertheless, the work is important and laudable. But maybe their claim that the work will support better exposure assessment for future epidemiological studies is too optimistic. Indeed, they could only demonstrate very few factors that affect dust concentrations significantly in that small set of households. And even for the significant findings the confidence limits remain very broad. Thus, the estimates are not very precise and maybe also not fully correct as the representativeness of the selected households for a larger population is not certain.

Line 45: the authors claim a “multi-year lag” in PUR reporting. I am not familiar with that Californian reporting system. But I see that they do report 2018 data based on PUR. Thus, the lag is not that bad! They apparently chose their study areas in 2019 based on 2018 data. And more generally, with rather low-level pesticide exposure in epidemiological studies I am not so much interested in acute and immediate effects. And for studying chronic effects we will always consider some time lag between exposure and outcome. So, for epidemiological studies, a reporting lag of a few years is not a severe problem!

Line 55: “…orchards (..) that are routinely applied”: I think the fields are not applied but “sprayed” (or treated with pesticides….)

Line 56: “these dynamics are also important environmental justice considerations.”: should it not read: “important FOR … considerations”?

Line 56: I have learned that “Latinx” is a new gender-neutral term for “Latino/Latina”. But why not use the old and gender neutral “Hispanic” instead?  

Figure 2: description says: “in California and study counties”. But the figure does not present total Californian data!

Line 168: “from 5 ng/g to 500 ng/ml”: you should use the same metric. I believe “ng/g” is correct, because in the next sentence you use “ppb”

Line 175: “For two samples, initial dust mass was not available,…”: I do not understand! You report mass per gram dust, not mass per sample. And for analysis you took aliquots or 100 mg each. (line 139) The original sample size is not relevant!

Lines 198ff: I do understand the cut-off at a p-value of 0.25. But the remaining part of the paragraph (ratio and variance) are over my head. Maybe it is my own ignorance only. But I would clearly be thankful if the authors could explain their reasoning in a way intelligible to me as well!

Figure 3: to the bare eye it seems that community 1 has higher total pesticide levels than any other community. I suppose it is only the lack of power (with only very few households per community) that left only the difference to the 2nd community significant.

Table 2: I do not understand the footnote. “Exposure variables” are Acephate, Bensulide, etc… What is the meaning of Diethyl phosphate (DEP); Diethyl dithiophosphate (DEDTP);… ? To the best of my knowledge, Acephate is not the same as Diethyl phosphate (I believe it is a dimethyl-phosphate!)

Abbreviations: Not “Ops”, but “OPs”. Not “Ppb”, but “ppb”!
You translate “OP” as “organophosphorus pesticides”, but in the text you also write “OP pesticides”. This indicates that OP stands for “organophosphorus” (or organophosphate) alone.
